# Do Advanced Spatial Strategies Depend on the Number of Flight Hours? The Case of Military Pilots

**DOI:** 10.3390/brainsci11070851

**Published:** 2021-06-25

**Authors:** Marco Giancola, Paola Verde, Luigi Cacciapuoti, Gregorio Angelino, Laura Piccardi, Alessia Bocchi, Massimiliano Palmiero, Raffaella Nori

**Affiliations:** 1Department of Biotechnological and Applied Clinical Sciences, University of L’Aquila, 67100 L’Aquila, Italy; marco.giancola@graduate.univaq.it (M.G.); massimiliano.palmiero@univaq.it (M.P.); 2Aerospace Medicine Department, Aerospace Test Division, Italian Air Force, 00071 Pomezia, Italy; luigi.cacciapuoti@aeronautica.difesa.it (L.C.); gregorio.angelino@aeronautica.difesa.it (G.A.); 3Department of Psychology “Sapienza”, University of Rome, 00185 Rome, Italy; laura.piccardi@uniroma1.it (L.P.); alessia.bocchi@gmail.com (A.B.); 4Cognitive and Motor Rehabilitation and Neuroimaging Unit, IRCCS Fondazione Santa Lucia, 00179 Rome, Italy; 5Department of Psychology, University of Bologna, 40126 Bologna, Italy; raffaella.nori@unibo.it

**Keywords:** cognitive style, spatial cognition, sense of direction, spatial orientation, mental rotation, individual differences

## Abstract

Background: Military pilots show advanced visuospatial skills. Previous studies demonstrate that they are better at mentally rotating a target, taking different perspectives, estimating distances and planning travel and have a topographic memory. Here, we compared navigational cognitive styles between military pilots and people without flight experience. Pilots were expected to be more survey-style users than nonpilots, showing more advanced navigational strategies. Method: A total of 106 military jet pilots from the Italian Air Force and 92 nonpilots from the general population matched for education with the pilots were enrolled to investigate group differences in navigational styles. The participants were asked to perform a reduced version of the Spatial Cognitive Style Test (SCST), consisting of six tasks that allow us to distinguish individuals in terms of landmark (people orient themselves by using a figurative memory for environmental objects), route (people use an egocentric representation of the space) and *survey* (people have a map-like representation of the space) user styles. Results: In line with our hypothesis, military pilots mainly adopt the survey style, whereas nonpilots mainly adopt the route style. In addition, pilots outperformed nonpilots in both the 3D Rotation Task and Map Description Task. Conclusions: Military flight expertise influences some aspects of spatial ability, leading to enhanced human navigation. However, it must be considered that they are a population whose navigational skills were already high at the time of selection at the academy before formal training began.

## 1. Introduction

Navigation in spatial surroundings is a cognitive process that requires prolonged maturation with the progression of skills, strategies and proficiency over the lifespan. Several theoretical frameworks have been proposed to explain how environmental knowledge is acquired and what the stages encompass. One of the first seminal theoretical models still used today is that by Siegel and White [1]. These authors hypothesized that environmental knowledge is acquired following three separate and hierarchical steps. The first is focused on *landmark knowledge* and is characterized by a figurative memory of environmental objects (i.e., buildings, fountains, shops, monuments). At this knowledge step, individuals may beacon towards a salient landmark, but they lack egocentric and allocentric information that would allow them to build up relationships between the individual and subsequent landmarks, as well as the relationship between landmarks along a path or absolute relationships between landmarks. The second step is *route knowledge* in which, through an egocentric perspective, the self-position allows individuals to determine the relation between landmarks met along the path. Finally, the last step is *survey knowledge*, in which a map-like representation is built, and through an allocentric perspective, mental environmental representation exists regardless of the self-position. When individuals reach this last step, they are able to find shortcuts and master metric knowledge of the environment itself. This model, although still adopted, is not free from criticism, and since 1990, a series of other theoretical models have been proposed to explain how the acquisition of environmental information and the creation of a representation of the surrounding world takes place [2,3,4]. Specifically, Tversky [2] suggests that for human beings, it is mandatory to take into account language; thus, to develop environmental knowledge, a further stage would be based on linguistic spatial categories. In contrast, Montello [3] disagrees with the hierarchical rigidity of the model, suggesting that *survey knowledge* can be achieved even after a fleeting exposure to an environmental map and that an individual does not necessarily have to pass through the three steps of Siegel and White’s model.

Undoubtedly, regardless of the theoretical model concerning environmental knowledge, several factors intervene in determining proficiency during spatial orientation. These factors can be internal and external. Among the external factors are environmental configuration, landmark visual accessibility, circulation systems and signage [5,6]. In contrast, internal factors include the individual’s inclination to capture some environmental information instead of other (field dependence/independence [7,8,9,10,11]), gender [12,13,14,15,16,17,18,19], age [20,21,22,23,24,25], familiarity with the environment and job-related expertise [4,26,27,28,29,30,31,32,33]. Internal factors can also include the navigational strategies that the individual prefers to use to navigate. Indeed, according to some authors, the three environmental knowledge steps proposed by Siegel and White correspond to three different strategies or spatial cognitive styles (SCSs) that individuals use when moving through the environment [34,35] regardless of their level of environmental knowledge. As a consequence, individuals may be categorized as *landmark-style* (LS), *route*-*style* (RS) and *survey-style* (SS) users, which correspond to the three different levels of navigational skills. *LS* users adopt less advanced and efficient navigational strategies than RS and *SS* users. Analyzing navigational behavior, *LS* users are poor navigators and very often experience the feeling of getting lost, while *RS* users are more skilled at estimating the place and time in which they have to turn right or left at a specific reference point from an egocentric perspective. Finally, *SS* users are good navigators; they have an external perspective, such as a bird’s-eye view, which allows for direct access to the global spatial layout [36], and are able to plan more flexible and efficient navigational strategies [7]. Moreover, Bocchi et al. [7] found that the navigational style also affects the sense of direction (SOD). People with the *survey style* have a better sense of direction than people with the *landmark* or *route styles* (e.g., [37]) and are more proficient in solving navigational problems and in travel planning [38,39].

Undoubtedly, most studies of spatial cognition have focused on internal factors, while external factors are of more interest to architects and geographers. Specifically, among internal factors, familiarity with the environment and job-related expertise deserve great attention because they are the most modifiable and least stable over the lifespan. For example, as familiarity and exposure time with the environment increase, women achieve navigational performance comparable to that of men [4,40]. Of course, familiarity and job-related experience are not precisely the same thing; while job-related experience allows one to generalize specific knowledge to similar situations, familiarity means that the person has only acquired high skills in that particular environment but is not able to transfer those abilities to other environments. Thus, the landmark user will remain a landmark user when approaching a new environment for the first time. In particular, navigational training that is similar to the experience gained at work has wider effects by producing jumps in the developmental stages that are previously acquired (e.g., [41]). This is noteworthy in terms of cognitive reserve and successful aging as well as in terms of the spillover of the effects on other skills. For example, it is known that mathematical disciplines are related to some aspects of spatial orientation [42,43] that could be potentiated through spatial cognition training. Among the internal factors, navigational strategies and cognitive styles play an important role because they can also influence other aspects of everyday life. In a recent work, Nori et al. [44] highlighted how mental survey representations result in fewer mistakes and infractions when driving a car. Similarly, Bocchi et al. [45] found that having a mental survey representation allows the implementation of an efficient strategy to search for a lost object. Generally, cognitive style refers to how people perceive the world and organize and process environmental information. Even cognitive style can influence the appreciation of a work of art (e.g., [46]). With respect to spatial orientation, individuals are able to grasp different aspects of the environment and consequently extrapolate certain spatial clues. It is a pervasive psychological dimension of individuals that is relatively stable throughout life, although it may adapt to environmental changes and pressures [47,48].

Most navigational skills require time to fully mature; these skills develop gradually and at distinct time points during childhood and early adolescence, even if spatial training may proficiently improve them [41,42,43,44,45,46,47,48,49], allowing some navigational milestones to be reached before normal maturity.

For some authors (e.g., [34,35]), navigational strategies correspond to actual cognitive styles that trace the stages of acquisition of environmental knowledge described by Siegel and White’s model, individuating three different styles—landmark, route and survey—with distinct competencies. Specifically, survey users are better at perspective-taking and spatial-orientation tasks, are good at finding their way back to a starting position along a path they experienced just once, are able to build more complex and flexible map-like representations and can study maps with a more efficient eye-movement pattern [50,51,52]. Undoubtedly, the literature on spatial orientation and military pilots highlights their higher levels of spatial skills than the general population. Indeed, to be able to fly a military jet, they need a better visuospatial working memory and attention and they must be very quick at mentally rotating 3D objects and processing environmental information, as well as making directional judgments [29,53]. Furthermore, they have a better topographic memory than nonpilots [28]. Gender differences are absent in this population, suggesting that the selection criteria for flight training are such that only individuals who have very good visual–spatial skills are admitted [28]. When asked through informal interviews about their navigational skills, military pilots spontaneously report that they believe they have always had a very good sense of direction, although they recognize that many people in the world are not so good at moving around in a new city or their own city. They also imagine that this ability is linked to the type of work they do, which has certainly contributed to increasing this ability, although studies in the literature do not always find a relationship between flight hours and navigational performance [28,29]. In this vein, Sutton et al. [54] also reported that even undergraduate student pilots were more accurate at estimating directions between landmarks in a virtual town than nonpilot controls. Furthermore, military pilots are better at recognizing 180° rotated objects than nonpilots when they have to estimate directional judgments about points learned from a different environmental perspective [29].

In this special population, the absence of gender differences in mental rotation and navigation would suggest the existence of advanced cognitive strategies and increased use of survey strategies that would partially explain their spatial skills. For example, Glicksohn and Naor-Ziv [55] found that pilots were more field independent than other populations, scored lower on neuroticism and scored higher on being experience-seeking. These authors also found a distinctive profile for military pilots relative to others who had served in combat units in the military.

To our knowledge, no studies have investigated which navigational style military pilots adopt during navigation. In several studies, Verde and coworkers [28,29,30] hypothesize that military pilots should be SS users taking into account their efficiency in mental rotation, topographic memory and perspective tasks. However, in their studies, only self-report measures were used; therefore, the authors could not draw any conclusions about the percentage of SS users in this population.

The aim of this study is to investigate whether military pilots have more advanced navigational strategies than the general population and whether these strategies have been developed as a consequence of their job-related training or already existed at the time of selection to enter the military academy. As a consequence, we formulate the following hypothesis: military pilots who are more skilled in cognitive processes underlying navigation adopt more survey navigational strategies than nonpilots, and the distribution of the three styles is significantly different from that in the nonpilot group. Indeed, in previous studies, military pilots self-reported being more prone to using the survey strategy than the route or landmark navigational strategies (see [28]). In the present study, we also explore the influence of flight hours on their performance in navigational tasks. For this purpose, we investigate whether, as the number of flight hours increase, their ability to perform spatial tasks also increases in terms of accuracy and timing or whether this ability is independent of related work experience. Indeed, it could be the consequence of a selection of certain cognitive characteristics that make an individual a good pilot.

## 2. Materials and Methods

### 2.1. Participants

One hundred ninety-eight healthy men (mean age = 29.22 years; SD = 6.94 years; age range = 19–50 years; mean education = 14.87 years; SD = 2.19 years; education range = 13–18 years) were enrolled in the experiment. The participants were divided into two groups: 92 nonpilots (mean age = 25.92; SD = 4.73; age range = 19–40; mean education = 14.57; SD = 2.16; education range = 13–18) from the general population without flight experience and 106 military jet pilots from the Italian Air Force (mean age = 32.08 years; SD = 7.30; age range = 20–50 years; mean education = 15.13 years; SD = 2.21 years; education range = 13–18 years) with the following flight experience: mean hours of flight = 724.11; SD = 1205.02 h; flight range = 15–6000 h. All participants signed a written informed consent form before undergoing the experiment. As indicated by the anamnestic questionnaire, none of the participants had a history of neurological or psychiatric disease. The pilots and nonpilots differed in terms of flight experience; the nonpilots did not have any flight experience and were matched for education (t_(196)_ = 1.463; *p* = 0.53) with the pilots. The Ethics Committee of the Department of Psychology of Bologna University (acceptance date: 14 March 2015), in agreement with the Declaration of Helsinki, approved this study.

### 2.2. Procedure

Before the experimental session, the participants were informed about the aim of the study, the procedure, their rights and the possibility of stopping the experiment at any time they chose. Afterward, the participants signed a written informed consent form and filled out a brief anamnestic questionnaire in which they had to indicate their age, educational level, addictions and state of health. Then, the Spatial Cognitive Style Test (SCST) [56] was administered individually to each participant of both groups in a quiet room.

### 2.3. Measures

The short version of the SCST used in this study [56] was designed to evaluate the three navigational cognitive styles (LS, RS and SS) used by people to move successfully through the environment. As pointed out by Nori and Giusberti [56], cluster analysis identified three different clusters in which the different spatial tasks were grouped: Cluster 1, Photo and Figure Task; Cluster 2, Sequence Task and Map Description Task; and Cluster 3, 3D Rotation Task and Sum and Straighten Task within Survey Tasks (for details see [56]). Based on the specific characteristics identified by each cluster, the authors considered Cluster 1 to be representative of the landmark strategy, Cluster 2 representative of the route strategy and Cluster 3 representative of the survey strategy. Seven items randomly presented composed every task, and for each subtest, the accuracy and the execution time scores were considered. No explicit instructions on execution time were given to the participants, who were only told to be accurate in performing the tasks. To record the execution time, a hand-held stopwatch was used, and the answers given by the participants were recorded on a grid. The subtest of the SCST, divided for navigational strategies, is described in detail below.

#### 2.3.1. Landmark Tasks

Photo Task. The participants were asked to study a photograph of a building for 3 s. Afterward, they had to recognize the building among photographs of four similar buildings (seven trials; see Figure 1A).

Figure Task. The participants had to study seven shapes for 75 s and were then asked to recognize them among 50 different figures, including the seven shapes previously studied (seven targets and 43 fillers; see Figure 1B).

#### 2.3.2. Route Tasks

Sequence Task. The participants were asked to study a photograph of an environmental scene from a first-person perspective for 15 s. Then, the participants presented the photograph divided into separate parts (3, 4 or 5 parts). The aim was to arrange the parts correctly, reconstructing the previously studied image (seven trials; Figure 1C).

Map Description Task. The participants were required to describe a pathway depicted on a map. Starting from a purple dot, the participants had to describe the route to reach a black dot by reporting the correct sequence of seven right–left turning points. Rotation of the map was explicitly required to perform the task (see Figure 1D).

#### 2.3.3. Survey Tasks

Three-Dimensional (3D) Rotation Task. The participants were asked to observe a picture of a TV on the left of an A4 paper. Afterward, they had to mentally rotate the shape in the direction indicated by one or two arrows following four possible rotations (90° to the left on the vertical axis, 90° to the right on the vertical axis, 90° from the top down on the horizontal axis or 90° from the ground upwards on the horizontal axis). Then, the participants had to choose the correct rotation among five possible pictures reported on the A4 paper (seven trials, see Figure 1E).

Sum and Straighten Task. The participants had to mentally sum and straighten a series of three segments depicted on an A4 paper to obtain the actual length and then indicate the correct answer among four alternatives (seven trials; Figure 1F).

Based on the cumulative nature of the Siegel and White model as described above and following the criteria of Nori and Giusberti [56,57], we computed the average of correct answers for each cognitive style with the purpose of obtaining a single score between 0 and 7 for each cognitive style. Based on this average score, we classified the participants as *LS* users if they provided at least an average score of ≥5 on the two *landmark* tasks and an average score of less than 5 on both the route and survey tasks. Participants who achieved at least a score of ≥5 in both the landmark and route tasks and less than an average score of 5 in the *survey* tasks were classified as *RS* users. Finally, participants who scored at least an average of ≥5 in the landmark, route and survey tasks were considered *SS* users. For two participants (one for each group), it was not possible to define the navigational style.

## 3. Results

To evaluate the distribution of the three navigational styles in both groups (pilots and nonpilots), Cochran’s Q test was performed. Regarding the pilot group, the variation among the three navigational styles (the frequencies among the three navigational styles are reported in Table 1) was significant (Cochran’s Q_2_ = 88.91; *p* = 0.000). More specifically, the comparisons were significant between LS and RS (Cochran’s Q_1_ = 26.00; *p* = 0.000), LS and SS (Cochran’s Q_1_ = 59.00; *p* = 0.000) and RS and SS (Cochran’s Q_1_ = 33.00; *p* = 0.000).

Regarding the nonpilot group, the variation among the three navigational styles (the frequencies among the three navigational styles are reported in Table 1) was significant (Cochran’s Q_2_ = 60.25; *p* = 0.000). More specifically, the comparisons between LS and RS (Cochran’s Q_1_ = 32.00; *p* = 0.001) and LS and SS (Cochran’s Q_1_ = 30.00; *p* = 0.000) were significant. The comparison between RS and SS was not significant (Cochran’s Q_1_ = 2.00; *p* = 0.16).

Three different Cochran’s Q tests were performed (one for each navigational style) to evaluate the distribution of the three navigational styles in both groups. The frequencies among the three navigational styles for pilots and nonpilots are reported in Table 1. Regarding LS, no differences were found between pilots and nonpilots (Cochran’s Q_1_ = 3.00; *p* = 0.083), whereas significant differences were found for both RS (Cochran’s Q_1_ = 9.00; *p* = 0.03) and SS (Cochran’s Q_1_ = 26.00; *p* = 0.000). Therefore, military pilots seem to show more complex navigational strategies than nonpilots.

Afterward, to evaluate the differences between pilots and nonpilots in each task of the SCST, we performed six separate one-way analyses of variance (ANOVAs) with group (pilots vs. nonpilots) as the independent variable and the accuracy scores of each task of the SCST (Photo Task, Figure Task, Sequence Task, Map Description Task, Three-Dimensional Rotation Task and Sum and Straighten Task) as the dependent variables. No differences between groups were found in the Photo Task (F_1,196_ = 1.51; *p* = 0.22; $\eta_{p}^{2}$ = 0.008), Figure Task (F_1,196_ = 0.23; *p* = 0.88; $\eta_{p}^{2}$ = 0.000), Sequence Task (F_1,196_ = 2.40; *p* = 0.12; $\eta_{p}^{2}$ = 0.012) or Sum and Straighten Task (F_1,196_ = 2.40; *p* = 0.12; $\eta_{p}^{2}$ = 0.012). However, we found that pilots outperformed nonpilots in the Map Description Task (F_1,196_ = 14.37; *p* = 0.00; $\eta_{p}^{2}$ = 0.068) and the Three-Dimensional Rotation Task (F_1,196_ = 11.56; *p* = 0.001; $\eta_{p}^{2}$ = 0.056) (see Figure 2).

Moreover, we performed six separate one-way analyses of variance (ANOVAs) with the execution time of every task of the SCST as the dependent variable and the group as the independent variable. Pilots seem to be slower than nonpilots in the Photo Task (F_1,196_ = 18.80; *p* = 0.00; $\eta_{p}^{2}$ = 0.088), the Figure Task (F_1,196_ = 12.95; *p* = 0.00; $\eta_{p}^{2}$ = 0.062), the Sequence Task (F_1,196_ = 12.73; *p* = 0.00; $\eta_{p}^{2}$ = 0.061) and the Map Description Task (F_1,196_ = 8.69; *p* = 0.004; $\eta_{p}^{2}$ = 0.042). No significant differences were found in the 3D Rotation Task (F_1,196_ = 2.95; *p* = 0.087; $\eta_{p}^{2}$ = 0.015) or the Sum and Straighten Task (F_1,196_ = 3.04; *p* = 0.083; $\eta_{p}^{2}$ = 0.015) (see Figure 3).

To evaluate whether the pilots’ expertise could influence their performance in the SCST, a correlation analysis was performed considering the flight hours and the accuracy score of each task of the SCST. No significant correlations were found. Descriptive statistics and Pearson’s correlations among the different variables are reported in Table 2.

Finally, to evaluate whether the pilots’ expertise could influence their performance in the SCST in terms of execution time, Pearson’s correlation analysis was performed considering the flight hours and the execution time of each task of the SCST. The analysis showed that the Map Description Task was positively correlated with flight hours (r = 0.247; *p* = 0.011). No significant correlations were found among the other tasks of the SCST and flight hours. Descriptive statistics and Pearson’s correlations among the different variables are reported in Table 3.

## 4. Discussion

In the present study, we investigated whether military pilots used more survey navigational strategies than nonpilots and whether flight hours may explain in part the use of advanced navigational strategies in this special population. In summary, we found a greater presence of survey users in the pilot group than in the nonpilot group. Furthermore, pilots outperformed nonpilots in specific tasks, such as the Map Description Task and Three-Dimensional Rotation Task. Considering the execution time, an unexpected result showed that pilots were slower than nonpilots in the Photo Task, Sequence Task and Map Description Task. We also observed an effect of flight hours only on the Map Description Task. It is known from the literature that spatial training may enhance performance significantly ([49] for a review), and this is true also with respect to the years of work experience. Indeed, Maguire and colleagues [58,59] showed that London taxi drivers had a larger hippocampus than other people doing different jobs as a result of their environmental knowledge, which was explicitly required even to obtain a license to drive a taxi. Our results showed that in the military pilot group, the distribution of the navigational styles is different from that in the control group with respect to route and survey style users. Generally, in the nonpilot population, most of the individuals use route navigational strategies to move through the environment, a few individuals are landmark users and, equally, a few individuals are survey users, which shows that there are a few people with little or no navigational skills and a few with excellent navigational skills. On the other hand, the ability to move around in the environment for humans is a skill that is learned in the very first months of life and perfected with practice. It is one of those skills in which a neurodevelopmental disorder (developmental topographical disorientation) that can affect healthy individuals who fail to develop adequate navigational strategies has been described [60,61]. On the other hand, there are also individuals who have strong navigational skills, including explorers, orienteers, forestry guides and undoubtedly pilots who, although they do not practice ground navigation, need to be able to use all those cognitive processes quickly, as they are at the basis of the ability to navigate successfully to fly. A military pilot needs to be fast and accurate when mentally rotating a target, has to learn a spatial configuration quickly, has to have accurate mental environmental representations [62] and, in general, must have useful cognitive resources to process spatial information. Interestingly, we also found that even if military pilots are already better at these skills when selected to enter the academy, certain aspects of navigation are influenced by flight hours; e.g., pilots with more flight hours were more accurate than others, even if they spent more time on the Map Description Task. Specifically, this task reflects their daily activities. Indeed, military pilots have to maintain cognitive coupling between two different reference frames corresponding to the map and the forward view of the world [14].

Mainly while carrying out their daily work activities, they use two kinds of reference frames: the egocentric reference frame (ERF), established by a pilot’s forward view out of the cockpit that directly corresponds to this perspective, and the world-centered reference frame (WRF), established by a visually presented map where the traditional canonical alignment is north-up [63]. Therefore, a pilot must be able to associate the current view of the world with its location on the map. In other words, [63] highlights that a pilot must be able to answer the question: ‘Am I (the ERF) where I should be (in the WRF)?’. As a consequence, the Map Description Task represents a very familiar activity, and it is well known that as familiarity increases, the ability to solve the task correctly increases [4,64].

However, it is noteworthy that, unlike everyday work activities, this task does not require speed but correctness, so they most likely opt for accurate performance at the expense of speed, which would explain, in our opinion, the result of the increased execution time compared to the general population, which is, however, more imprecise. Therefore, even in the case of military pilots, their work experience affects some specific navigational skills that increase over time. SS users have a map-like representation, which implies the use of an allocentric perspective that requires a mental representation of the environment regardless of the individual’s position, that is, a better ability to reach a planned goal, to find novel paths and shortcuts when the familiar route is not accessible and to have metric knowledge. It is also known that the relationship with the environment is reciprocal because mental representations are modified whenever an environmental change occurs. A peculiar case that has been studied is the mental reorganization that an individual is forced to perform when the environment is completely changed due to a natural disaster. In that case, the individual is exposed to a kind of intensive navigational training that leads him or her to relearn the environment. This is the case, for example, described by Piccardi et al. [65], who found that individuals without psychological disorders who were exposed to the L’Aquila earthquake had an increase in topographic memory capability resulting from the need to relearn the configuration of the surrounding environment. These data are in line with the fact that although pilots already have a propensity for spatial orientation, as flight hours increase, some crucial navigational skills improve, at least in terms of the speed of execution. The presence in our sample of more SS users supports data from past studies in which military pilots self-reported using cardinal points to orient themselves, visualizing a path not from a first-person perspective but like a map and thinking about distances in meters. Our results are also in line with those found by Glicksohn and Naor-Ziv [55], in which they observed the presence of more field-independent individuals in military populations than in other populations. Considering the single tasks of the SCST, we found that military pilots outperformed nonpilots in two tasks: the Map Description Task and the Three-Dimensional Rotation Task. In both, pilots are more accurate than nonpilots. The finding that pilots did not differ from nonpilots in the Sum and Straighten Task, which is one of the tasks for measuring survey strategies, deserves separate mention. Actually, it is the most difficult task because it requires the subject to simultaneously keep in mind different components of the spatial structure and manipulate them by processing them later. The absence of differences in performing the task between the two groups can be interpreted as an effect of the high cognitive load requirement, that is, the amount of working memory resources used [66]. In fact, cognitive load theory differentiates load into three types: (i) intrinsic cognitive load, which concerns the intrinsic complexity of information that must be understood and the material that must be learned; (ii) extraneous cognitive load, which concerns how instructions are provided and may be imposed by instructions that are less than optimal (for instance, nonoptimal instructional procedures are considered to impose an extraneous cognitive load); and (iii) germane cognitive load, which concerns the acquisition of knowledge and refers to learner characteristics [67]. Specifically, the Sum and Straighten Task requires a different level of intrinsic cognitive load [68,69].

When the execution times are considered, pilots appear slower than nonpilots in the Photo Task, the Figure Task, the Sequence Task and the Map Description Task. It is important to point out that although the execution times were recorded, the subjects were never told to aim for speed, but rather to aim for accuracy. This could explain the slowness of execution of some tasks in the group of pilots; in the absence of an explicit order of speed, they preferred greater accuracy of execution. In fact, when they were specifically told to be fast, their performance outperformed that of the general population. Verde et al. [53] found that pilots are much faster than nonpilots in two-dimensional rotations. In contrast, in the present study, we found no difference with nonpilots in terms of speed in three-dimensional rotations, but the instructions provided in the two studies were different; whereas in the study of Verde et al. [53], pilots were explicitly told to be accurate and fast, in our study they were only told to be accurate. The present result is also in line with the data presented by Verde et al. [29] with respect to directional judgments, where pilots are slower in providing directional judgments but are much more accurate even when judgments are counter-aligned.

Undoubtedly, maintaining orientation during flight requires rotating in three dimensions with a higher workload than spatial orientation in ground navigation, and it is surely an even more critical skill for a flight expert. A military pilot has to have *navigational awareness*, which requires establishing the geometries between egocentric and allocentric systems and would explain their ability to mentally represent the environment like a map without considering their own position. To reach such awareness, mental rotation ability, triangulation, image comparison and translation represent the four fundamental cognitive operations necessary during flight [63]. Considering our data, we can state that the flight expertise and specific flight training practiced by pilots would not seem to completely influence the type of ground navigational style because flight hours did not correlate with all navigational tasks. Certainly, pilots self-perceived themselves to have a good sense of direction in the ground navigation context (see [28]), and indeed, they performed better than nonpilots in several tasks characterizing survey users. Surely, an advanced spatial competence characterizes this special population, and it is certainly thanks to their spatial skills that they were able to pass the selection requirements and enter the academy. Our data seem to confirm what had emerged in some of our previous studies—that professional experience has a minimal effect on the skills already present in this population. Therefore, in the case of military pilots, it would be a selection bias to group together individuals who were already endowed with high visuospatial skills. However, it is fair to point out the presence of some limitations. The first is not having a perfectly matched sample by age but only by the level of education; although the age difference between the two groups was not large, the group of pilots was significantly older than the group of nonpilots. This, from a certain point of view, makes their abilities even more evident because it is known in the literature that visuospatial abilities are among the most sensitive to the effects of age (i.e., [70]). Furthermore, it is known that age has a reduced influence in regard to understanding the weight of expertise. Indeed, Horton et al. [71] point out that experts are observed as having a reduced effect of age along the age-decline curve of cognitive processes. Therefore, if the pilot group continues to have significantly better navigational strategies, it means that, in their case, the effects of age are reduced, which could be explained partly by their expertise, assuming that expertise acts as a protective factor. A second limitation may be the decision to select only men and not women pilots in the study. Gender differences in visuospatial abilities are well described (i.e., [72]), and such differences do not emerge in the pilot group (e.g., [28,53]), in which men and women perform visuospatial tasks equally well. This decision was made because the number of women in the Italian Air Force is still small and the final sample would have been strongly unbalanced by gender. Undoubtedly, future studies should also investigate the navigational strategies used by female pilots with respect to male pilots and the general population. Finally, it should also be taken into account that a large number of correlations might not lead to highly reliable results; as a consequence, the correlations between flight hours and the six tasks should be interpreted with caution. However, the only significant correlation was theoretically sensible according to our hypothesis.

Overall, despite these limitations, the present study sheds some light on the navigational strategies preferred by pilots in ground navigation, confirming that their advanced spatial skills were very likely already developed at the time of selection to enter the academy and were continuously improved while executing their job duties.

## 5. Conclusions

In summary, our study highlights the presence of advanced spatial abilities and navigational strategies in military pilots, showing that they are mainly survey-style users. On the other hand, flight hours influence only one of the navigational tasks, the Map Description Task, which is a task more often performed by pilots in their daily activities. This further confirms that familiarity with a task contributes to improving navigational competence.

## Figures and Tables

**Figure 1 brainsci-11-00851-f001:**
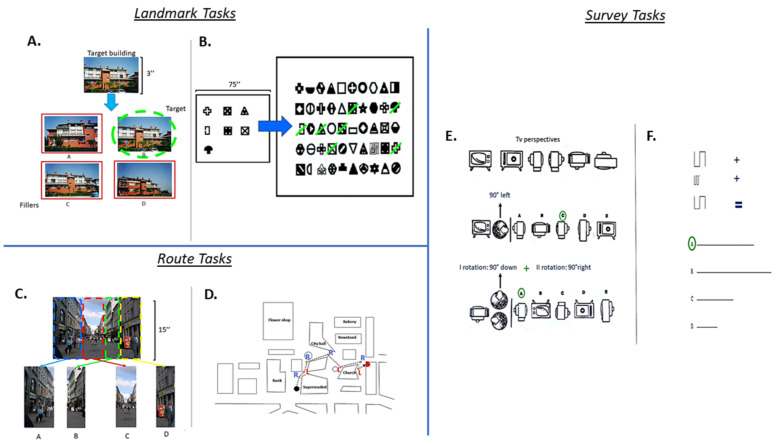
An example of each type of task used to measure navigational strategies (landmark, route and survey) is reported. (**A**). Example of Photo Task; (**B**). Figure Task; (**C**) Example of Sequence Task; (**D**). Map Description Task; (**E**). Example of 3D Rotation Task; (**F**). Example of Sum and Straighten Task.

**Figure 2 brainsci-11-00851-f002:**
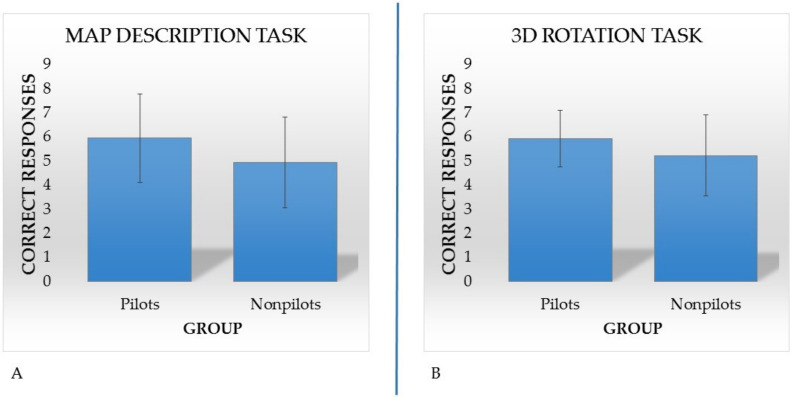
(**A**) Mean plot: In the x-axis, the two groups (pilots and nonpilots) are reported. The y-axis shows the participants’ mean accuracy score on the Map Description Task of the Spatial Cognitive Style Test (SCST). (**B**) Mean plot: In the x-axis, the two groups (pilots and nonpilots) are reported. The y-axis shows the participants’ mean accuracy score on the 3D Rotation Task of the SCST.

**Figure 3 brainsci-11-00851-f003:**
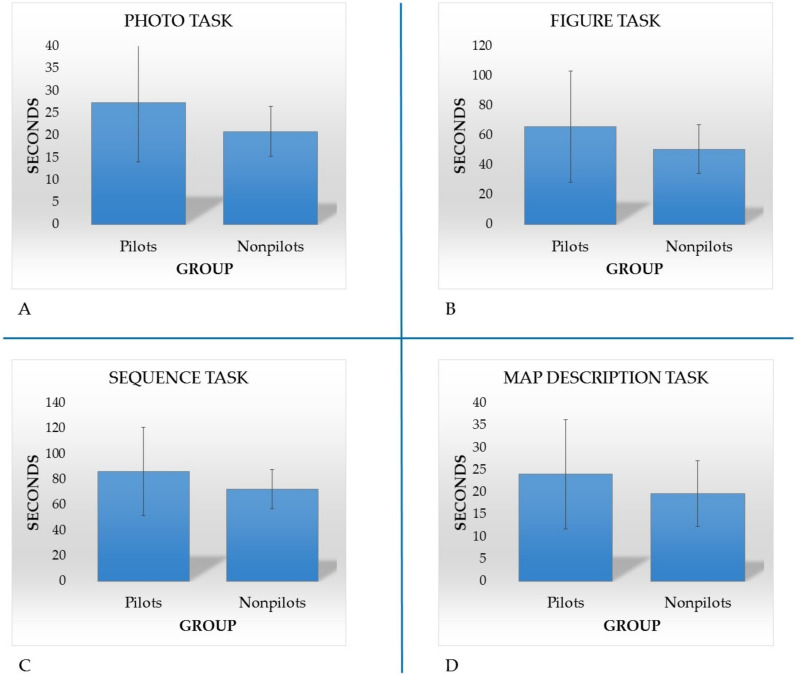
Mean plot: In the x-axes, the two groups (pilots and nonpilots) are reported. The y-axes show the participants’ mean execution time (s) on the Photo Task (**A**), Figure Task (**B**), Sequence Task (**C**) and Map Description Task (**D**) of the SCST.

**Table 1 brainsci-11-00851-t001:** Frequencies of the three cognitive styles in the pilot and nonpilot groups.

	Pilot Groups		Nonpilot Groups	
Cognitive Style	Frequencies	Percentages	Frequencies	Percentages
Landmark	7	6.61%	10	10.87%
Route	33	31.13%	42	45.65%
Survey	66	62.26%	40	43.48%

**Table 2 brainsci-11-00851-t002:** Descriptive statistics and Pearson’s correlations among accuracy scores of the SCST tasks and flight hours.

	Min	Max	M	SD	1	2	3	4	5	6	7
SCST-Photo Task (1)	5.00	7.00	6.80	0.42	1	0.05	0.08	0.06	0.03	0.01	0.09
SCST-Figure Task (2)	4.00	7.00	6.56	0.68	0.05	1	0.08	−0.03	0.07	0.08	−0.07
SCST-Sequence Task (3)	2.00	7.00	6.19	1.08	0.08	0.08	1	0.04	0.36	0.10	0.02
SCST-Map Description Task (4)	2.00	7.00	5.96	1.83	0.06	−0.03	0.04	1	0.17	0.03	−0.09
SCST-3D Rotation Task (5)	2.00	7.00	5.93	1.17	0.03	0.07	0.36	0.17	1	0.006	0.01
SCST-Sum and Straighten Task (6)	0.00	6.00	3.12	1.20	0.1	0.08	0.10	0.03	0.00	1	−0.06
Flight Hours (7)	15.00	6000.00	724.11	1205.02	0.09	−0.07	0.02	−0.09	0.01	−0.064	1

**Table 3 brainsci-11-00851-t003:** Descriptive statistics and Pearson’s correlations among execution time scores of the SCST tasks and flight hours. The correlation of interest is reported in bold.

	Min	Max	M	SD	1	2	3	4	5	6	7
SCST-Photo Task (1)	10.01	86.03	27.40	13.36	1	0.28	0.45	0.19	0.14	0.14	−0.02
SCST-Figure Task (2)	18.00	260.00	65.95	37.35	0.28	1	0.180	0.13	0.18	0.10	−0.07
SCST-Sequence Task (3)	34.00	284.57	86.24	34.59	0.45	0.18	1	0.38	0.37	0.36	0.04
SCST-Map Description Task (4)	5.66	64.00	24.01	12.30	0.19	0.13	0.38	1	0.32	0.15	**0.24**
SCST-3D Rotation Task (5)	23.00	200.94	76.60	35.40	0.14	0.18	0.37	0.32	1	0.34	−0.15
SCST-Sum and Straighten Task (6)	27.51	411.56	95.52	56.41	0.14	0.10	0.36	0.15	0.34	1	−0.11
Flight Hours (7)	15.00	6000.00	724.11	1205.02	−0.02	−0.07	0.04	0.24	−0.15	−0.11	1

## Data Availability

The data presented in this study are available on request from the corresponding author. The data are not publicly available due to subject confidentiality.

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
