# Peer review of "Do Advanced Spatial Strategies Depend on the Number of Flight Hours? The Case of Military Pilots"

_brainsci, 2021, doi:10.3390/brainsci11070851_

Round 1

Reviewer 1 Report

The authors investigated spatial abilities in military pilots and compared this group with a sample of non-pilots. All participants performed a reduced version of the Spatial Cognitive Style Test (SCST). As hypothesized military pilots outperformed non-pilots in accuracy in the Three-Dimensional Rotation Task, which is described as a survey strategy task. Furthermore, pilots were more accurate in the Map Description Task, which is described as a route strategy task. Analyses of the response time showed that pilots were slower than non-pilots in all landmark and route strategy tasks, but not in both survey strategy tasks. No correlations were observed between the number of flight hours and task accuracy. Flight hours were positively correlated with response times on the Map Description Task. The authors conclude that military pilots use a survey strategy when navigating and that this advanced spatial ability already exists prior to professional fight training. This is an interesting paper, but some parts need revision.

Major comments:

  1. The authors claim in the title and at the end of the introduction that they investigate whether spatial abilities in pilots are acquired or innate (title and sentence in lines 170 to 173). With the present study design, it is impossible to test innateness. Even though no correlation with flight hours is observed and pilots self-reported that they experienced advanced spatial skills already early in life, it cannot be concluded that the spatial skill was present at birth. I also think it is highly unlikely that such a complex skill like using a survey strategy can be present without development during childhood. I think the innateness claim should be removed from the title and the introduction. The authors might want to include the topic about the development of strategies and possible innateness as a speculative suggestion in the discussion.
  2. I would also change ‘high level of navigational strategy’ in the title and would rather talk about the survey strategy as an advanced spatial strategy.
  3. The discussion would benefit from a clearer structure. It would be better to start the Discussion with a short summary of the main findings and then discuss every result in detail. The finding that pilots outperform non-pilots in the Map Description task, which is a route strategy task, is not explicitly discussed. Also, why pilots perform similarly to non-pilots in the Sum and Straighten Task, which is the second survey task, is not discussed.
  4. It would be better to combine Table 1 and 2. Table 3 gives double information with the frequencies and should be removed.

Minor comments:

There are a number of errors, typos and unclear points in the manuscript. The authors should carefully check the manuscript, especially the introduction and discussion contain errors.

Lines 46/48: It would be better to replace ‘advanced’ with proposed or introduced here.

Line 85: ‘Landmark style users are less able than route and survey style users.’ It is unclear here what is meant with being less able.

Line 96: ‘There is no doubt that the most of studies of spatial cognition…’. Rephrase here ‘…that most of the studies on spatial cognition…’

Line 99: ‘…deserves..’ should be without s.

Line 101: ‘…women who are generally less able than men..’. I would not generally state this, the story is more complex.

Line 137: ‘…skills than general population.’ should be …than in the general population.

Line 166: ‘…taking into account for their…’ here ‘for’ should be removed.

Line 171: ‘…strategies than general population…’ should be …than in the general population.

Line 180: ‘space tasks’ should be ‘spatial tasks’

Line 178 – 183: This is a very long sentence. It is better to rephrase this.

Lines 186-194: I guess the authors report standard deviations in the participant section. It would be better to include SD then.

Line 249: ‘tasks’ should be without s

Lines 309/310: ‘It is known in literature that spatial training may enhance significantly performances ([49] for a review) and…’ I would rephrase here: ‘It is known from the literature that spatial training may enhance performance significantly …’

Lines 309-314: This is a very long sentence. I would start a new sentence with Maguire.

Line 312: ‘…than other categories…’ this is unclear.

Line 316: ‘…in the non-pilot population the most of individuals…’. This should be ‘…in the non-pilot population most of the individuals…’

Line 329: ‘…he has to learn fast a spatial configuration…’. Better would be ‘…he/she has to learn a spatial configuration quickly…’

Lines 333/334: ‘…more time in Map Description task…’. This should be ‘…more time on the Map Description task…’

Lines 334/335: ‘So also in the case of pilots, the effects of the work they do affect their skills, which increase over time.’ I would rephrase the sentence.

Lines 335 – 337: I would rephrase the sentence, some of the information seems to be double here.

Lines 386/387: The authors write about the spatial attitude of pilots here. I find ‘attitude’ confusing here. I would discuss ‘spatial ability’.

Line 390-392: I find this last sentence unclear. It would be better to rephrase this.

Author Response

All changes in the manuscript are highlighted in yellow colour

Reviewer 1

The authors investigated spatial abilities in military pilots and compared this group with a sample of non-pilots. All participants performed a reduced version of the Spatial Cognitive Style Test (SCST). As hypothesized military pilots outperformed non-pilots in accuracy in the Three-Dimensional Rotation Task, which is described as a survey strategy task. Furthermore, pilots were more accurate in the Map Description Task, which is described as a route strategy task. Analyses of the response time showed that pilots were slower than non-pilots in all landmark and route strategy tasks, but not in both survey strategy tasks. No correlations were observed between the number of flight hours and task accuracy. Flight hours were positively correlated with response times on the Map Description Task. The authors conclude that military pilots use a survey strategy when navigating and that this advanced spatial ability already exists prior to professional fight training. This is an interesting paper, but some parts need revision.

Major comments:

  1. The authors claim in the title and at the end of the introduction that they investigate whether spatial abilities in pilots are acquired or innate (title and sentence in lines 170 to 173). With the present study design, it is impossible to test innateness. Even though no correlation with flight hours is observed and pilots self-reported that they experienced advanced spatial skills already early in life, it cannot be concluded that the spatial skill was present at birth. I also think it is highly unlikely that such a complex skill like using a survey strategy can be present without development during childhood. I think the innateness claim should be removed from the title and the introduction. The authors might want to include the topic about the development of strategies and possible innateness as a speculative suggestion in the discussion.

Reply: We thank the Reviewer for his/her comment that we considered removing from both the title and the introduction the term innate and innatism and using it in a different form as speculation in the discussion.

  1. I would also change ‘high level of navigational strategy’ in the title and would rather talk about the survey strategy as an advanced spatial strategy.

Reply: Done. We modified title as suggested.

  1. The discussion would benefit from a clearer structure. It would be better to start the Discussion with a short summary of the main findings and then discuss every result in detail. The finding that pilots outperform non-pilots in the Map Description task, which is a route strategy task, is not explicitly discussed. Also, why pilots perform similarly to non-pilots in the Sum and Straighten Task, which is the second survey task, is not discussed.

Reply: We thank Reviewer for his/her suggestion, following them we introduce discussion with a short summery of results and then we discussed single results. Concerning the military pilots’ performance on Map Description Task we interpreted this result in the light of their work activities quoting Aretz who describes very well the interaction between the two frames of reference, the egocentric and the allocentric one, that allow the pilot to know his position and to use the map, this would explain their higher performance in an egocentric task. With respect to the increase in the execution time, we interpret this as a direct consequence of the instructions of the task where it was not explicitly asked to be fast. As already found in the literature (e.g., Verde et al. 2018 “Spatial orientation and directional judgments in pilots vs. nonpilots

Aerospace Medicine and Human Performance, 2018, 89(10), pp. 857–862”) when this instruction is missing pilots prefer accuracy to speed.

We now also comment on the absence of significant differences between pilots and non-pilots in the Sum and Straighten task, explaining this result in light of the fact that this task is the most difficult of the survey tasks and therefore requires a high cognitive load. On the basis of the increased cognitive load hypothesis, we discussed the finding (page 15 lines 412-428).

  1. It would be better to combine Table 1 and 2. Table 3 gives double information with the frequencies and should be removed.

Reply: Done

Minor comments:

There are a number of errors, typos and unclear points in the manuscript. The authors should carefully check the manuscript, especially the introduction and discussion contain errors.

Lines 46/48: It would be better to replace ‘advanced’ with proposed or introduced here. Okay

Line 85: ‘Landmark style users are less able than route and survey style users.’ It is unclear here what is meant with being less able.

Reply: We clarified the sentence as follows: Landmark style users adopt less advanced and efficient navigational strategies than the route and survey style users.

Line 96: ‘There is no doubt that the most of studies of spatial cognition…’. Rephrase here ‘…that most of the studies on spatial cognition…’ DONE

Line 99: ‘…deserves..’ should be without s.  OK

Line 101: ‘…women who are generally less able than men..’. I would not generally state this, the story is more complex.

Reply: We deleted this part of the sentence and wrote like this: women achieve navigational performance comparable to that of men

Line 137: ‘…skills than general population.’ should be …than in the general population. ok

Line 166: ‘…taking into account for their…’ here ‘for’ should be removed.OK

Line 171: ‘…strategies than general population…’ should be …than in the general population. OK

Line 180: ‘space tasks’ should be ‘spatial tasks’ OK

Line 178 – 183: This is a very long sentence. It is better to rephrase this. DONE

Lines 186-194: I guess the authors report standard deviations in the participant section. It would be better to include SD then. DONE

Line 249: ‘tasks’ should be without s OK

Lines 309/310: ‘It is known in literature that spatial training may enhance significantly performances ([49] for a review) and…’ I would rephrase here: ‘It is known from the literature that spatial training may enhance performance significantly …’

Reply: We rephrased the sentence

Lines 309-314: This is a very long sentence. I would start a new sentence with Maguire. DONE

Line 312: ‘…than other categories…’ this is unclear.

Reply: We clarified the sentence

Line 316: ‘…in the non-pilot population the most of individuals…’. This should be ‘…in the non-pilot population most of the individuals…’ OK

Line 329: ‘…he has to learn fast a spatial configuration…’. Better would be ‘…he/she has to learn a spatial configuration quickly…’ OK

Lines 333/334: ‘…more time in Map Description task…’. This should be ‘…more time on the Map Description task…’ OK

Lines 334/335: ‘So also in the case of pilots, the effects of the work they do affect their skills, which increase over time.’ I would rephrase the sentence.

We rephrased as follows: “Therefore, even in the case of military pilots, their work has an effect on their navigational skills which increases over time.”

Lines 335 – 337: I would rephrase the sentence, some of the information seems to be double here.

Reply: We rephrased as follows: “Survey users have a map-like representation, which implies the use of an allocentric perspective that requires a mental representation of the environment regardless the individual’s position.”

Lines 386/387: The authors write about the spatial attitude of pilots here. I find ‘attitude’ confusing here. I would discuss ‘spatial ability’.

Reply: We deletend attitude along the text.

Line 390-392: I find this last sentence unclear. It would be better to rephrase this.

Reply: We rephrased the sentence as follows: “Considering our data, we can state that the flight expertise and specific flight training practiced by pilots would not seem to completely influence the type of ground navigational style, because flight hours did not correlate with all navigational tasks. Certainly, pilots self-perceived themselves as people with a good sense of direction…”

Reply. We thank the Reviewer for his/her careful and accurate reading and for pointing out typos in the work which we have corrected in this revised version. In addition, we have sent the work to AJE who have done a certified English editing.

Reviewer 2

The paper presents an interesting body of data. However, I have a number of reservations

  1. It is not clear how the tasks can be assigned to the three levels (landmark, route, survey). For example, the 3D rotation task is really just a mental rotation with out any memory involvement. Why does high score in this task relate to survey navigation. The sum and straighten task can be seen as a path integration over an imagined travel along the marked lines. Again, it does not invovle memory. Finally, what is landmark-specific about the landmark tasks, they are just object recognition.

Reply:  With respect to the tasks of the SCST (Nori & Giusberti, 2006 “American Journal of PsychologyVolume 119, Issue 1, Pages 67 – 86 Spring  Predicting cognitive styles from spatial abilities”) and what they measure it is noteworthy to consider that SCST is a standardised battery used to assess navigational strategies already considered from other groups with this intent (e.g., Navigation ability test: A new specific test to asses spatial orientation ability in football players and healthy subjects; Gamba, P., Guidetti, R., Guidetti, G.     2020    Journal of Sports Medicine and Physical Fitness

60(6), pp. 934-941; Using Web-Based GIS to Assess Students' Geospatial Knowledge of Hurricanes and Spatial Habits of Mind  Perugini, S., Bodzin, A.M.           2020    Journal of Geography

119(2), pp. 63-73; Digital Map Design Elements for Local Tourism: Comparing User Cognition Between Age of 20s and Above 60      Li, M.L., Chen, M.S., Sato, K.           2020    Advances in Intelligent Systems and Computing 1256 AISC, pp. 55-65; Effects of Task Demand and Familiarity with Scenes in Visuospatial Representations on the Perception and Processing of Spatial Information           Kim, K., Kim, M. 2018 Journal of Geography 117(5), pp. 193-204). Specifically, the 3D Mental Rotation Task and Sums and Straighten Task in reality require the use of working memory, because in the 3D Mental Rotation Task the subject is asked to perform a double mental rotation, which in order to be done requires keeping the target online in mind and its active processing and manipulation in order to correctly choose the different alternatives present. Compared to Sums and Straightens Task, the task is not only a path integration task that requires to sum the vectors of distance and direction travelled from a starting point to estimate current position, and so the path back to the starting point, but requires otherwise to manipulate spatial segments in order to imagine them in a final configuration that is never present on paper and therefore again requires to keep in mind and manipulate more information and also involves the evaluation of a Euclidean distance that is a survey type competence. This task is also the most cognitively demanding requiring the higher cognitive load to solve it. Indeed, this task is difficult for both groups in spite of job-related expertise and advanced navigational strategies.

With respect to landmark tasks they evaluate the recognition of an object regardless of its position in space and the individual's position in space. It is in fact the poorest navigational strategy and does not require spatial skills but only visual ones. In reality, the proposed tasks require, from one side, the recognition of environmental objects such as landmarks, in fact the Photo Task includes only photos of buildings and from the other side, the other task requires the recognition of abstract symbols. All tasks included in the SCST have been developed within the Siegel and White’s Model that distinguishes landmark, route and survey competences such as: landmark knowledge à characterized by a figurative memory of environmental objects (i.e., buildings, fountains, shops, monuments); route knowledge à the self-position allows to determine the relation between landmarks met along the path; and survey knowledge à the mental environmental representation exists in spite of the self-position.

This Model of environmental knowledge remains to date the most cited model for the development of navigational skills (1155 on Scopus).

  1. As far as I understand, a subject scoring 7 in the route tasks and 5 in the "survey tasks" would be classified as a survey user. Why? besides, 80% of 7 is 5.6, which is more than 5.

Reply: We would like to thank the Reviewer for his/her comment, which allowed us to note an important typo concerning the way in which participants were classified, the percentage of which does not correspond to 80%, and we have now rewritten the part on the classification of styles in the hope that it will be clearer.

  1. Figure 1 is important but had very low quality in my copy. this must be improved.

Reply: We improved the quality of the figure.

  1. Table 3 is redundant and should be deleted.

Reply: Following also the suggestions of Reviewer 1 we collapsed the three tables in only 1 table, therefore table 3 is not more in the manuscript.

  1. The paper can be substantially improved by showing a correlation matix between the xis tasks. if the three strategies can indeed be distinguished, one would expect to the three 2x2 submatrices of high correlation along the diadonal. This might even be done in a factor analysis.

Reply:In the original study of 2006 on the standardization of the SCST battery, through a cluster analysis the three different factors that allowed to classify the tasks within the three cognitive styles had already been extrapolated. This is now added in the instruments section. In our eyes computing a correlation matrix between the six tasks could be not the right procedure, because we have a sample that is not normal buti t is characterized by advanced navigational strategies, indeed in the military pilots sample the frequency of SS users is higher than in general population. On the other hand if we ruled out military pilots from the total sample, the sample is too small for computing the correlation matrix only among the SCST’s tasks.

  1. The increased reaction time in the skilled pilots is surprisong. is there a good explanation of this?

Reply: we interpreted the increase in response times in military pilots as an effect of the fact that the instruction required accuracy but not speed, so this group prefers to be accurate rather than fast, otherwise when asked to be fast and accurate their performance is significantly faster and more accurate than controls as in the article by Verde and collaborators (2013). Differently, when pilots did not receive such instruction they appear more accurate but slower as in Verde et al 2018. We have explicitly included this explanation in the article in the Discussion section.

  1. The title is missleading. I found no data addressing the innateness problem.

Reply: We deleted from the title the innateness problem as well as along the text.

Reviewer 2 Report

The paper presents an interesting body of data. However, I have a number of reservations

  1. It is not clear how the tasks can be assigned to the three levels (landmark, route, survey). For example, the 3D rotation task is really just a mental rotation with out any memory involvement. Why does high score in this task relate to survey navigation. The sum and straighten task can be seen as a path integration over an imagined travel along the marked lines. Again, it does not invovle memory. Finally, what is landmark-specific about the landmark tasks, they are just object recognition.
  2. As far as I understand, a subject scoring 7 in the route tasks and 5 in the "survey tasks" would be classified as a survey user. Why? besides, 80% of 7 is 5.6, which is more than 5.
  3. Figure 1 is important but had very low quality in my copy. this must be improved.
  4. Table 3 is redundant and should be deleted.
  5. The paper can be substantially improved by showing a correlation matix between the xis tasks. if the three strategies can indeed be distinguished, one would expect to the three 2x2 submatrices of high correlation along the diadonal. This might even be done in a factor analysis.
  6. The increased reaction time in the skilled pilots is surprisong. is there a good explanation of this?
  7. The title is missleading. I found no data addressing the innateness problem.

Author Response

(The authors gave the same response as above.)
